# Comparative oncogenomics identifies combinations of driver genes and drug targets in *BRCA1*-mutated breast cancer

Stefano Annunziato[1,2,3], Julian R. de Ruiter[1,2,3,4], Linda Henneman[1,5], Chiara S. Brambillasca[1,2,3], Catrin Lutz[1,2,3], François Vaillant [6,7], Federica Ferrante[1,2,3], Anne Paulien Drenth[1,2,3], Eline van der Burg[1,2,3], Bjørn Siteur[8], Bas van Gerwen[8], Roebi de Bruijn[1,2,3,4], Martine H. van Miltenburg[1,2,3], Ivo J. Huijbers[5], Marieke van de Ven[8], Jane E. Visvader [6,7], Geoffrey J. Lindeman [6,9,10], Lodewyk F.A. Wessels[2,3,4] & Jos Jonkers[1,2,3]

*BRCA1*-mutated breast cancer is primarily driven by DNA copy-number alterations (CNAs) containing large numbers of candidate driver genes. Validation of these candidates requires novel approaches for high-throughput in vivo perturbation of gene function. Here we develop genetically engineered mouse models (GEMMs) of BRCA1-deficient breast cancer that permit rapid introduction of putative drivers by either retargeting of GEMM-derived embryonic stem cells, lentivirus-mediated somatic overexpression or in situ CRISPR/Cas9-mediated gene disruption. We use these approaches to validate *Myc*, *Met*, *Pten* and *Rb1* as *bona fide* drivers in BRCA1-associated mammary tumorigenesis. Iterative mouse modeling and comparative oncogenomics analysis show that MYC-overexpression strongly reshapes the CNA landscape of BRCA1-deficient mammary tumors and identify MCL1 as a collaborating driver in these tumors. Moreover, MCL1 inhibition potentiates the in vivo efficacy of PARP inhibition (PARPi), underscoring the therapeutic potential of this combination for treatment of *BRCA1*-mutated cancer patients with poor response to PARPi monotherapy.

[1] Division of Molecular Pathology, The Netherlands Cancer Institute, 1066 CX Amsterdam, The Netherlands. [2] Cancer Genomics Netherlands, The Netherlands Cancer Institute, 1066 CX Amsterdam, The Netherlands. [3] Oncode Institute, The Netherlands Cancer Institute, 1066 CX Amsterdam, The Netherlands. [4] Division of Molecular Carcinogenesis, The Netherlands Cancer Institute, 1066 CX Amsterdam, The Netherlands. [5] Transgenic Core Facility, Mouse Clinic for Cancer and Aging (MCCA), The Netherlands Cancer Institute, 1066 CX Amsterdam, The Netherlands. [6] ACRF Stem Cells and Cancer Division, Walter and Eliza Hall Institute of Medical Research, Parkville, VIC 3052, Australia. [7] Department of Medical Biology, University of Melbourne, Parkville, VIC 3010, Australia. [8] Preclinical Intervention Unit, Mouse Clinic for Cancer and Aging (MCCA), The Netherlands Cancer Institute, 1066 CX Amsterdam, The Netherlands. [9] Department of Medicine, University of Medicine, Parkville, VIC 3010, Australia. [10] Parkville Familial Cancer Centre, Royal Melbourne Hospital and Peter MacCallum Cancer Centre, Parkville, VIC 3050, Australia. These authors contributed equally: Stefano Annunziato, Julian R. de Ruiter, Linda Henneman, Chiara S. Brambillasca. These authors jointly supervised this work: Lodewyk F. A. Wessels, Jos Jonkers. Correspondence and requests for materials should be addressed to L.F.A.W. (email: l.wessels@nki.nl) or to J.J. (email: j.jonkers@nki.nl)

Triple-negative breast cancer (TNBC) accounts for 10–15% of all breast cancers and is characterized by lack of expression of the estrogen receptor (ER), the progesterone receptor (PR), and the human epidermal growth factor receptor 2 (HER2). Due to the lack of these receptors, TNBCs cannot be treated with targeted therapies that have been effective in treating other breast cancer subtypes. As a result, TNBC has a relatively poor clinical prognosis and chemotherapy remains its current standard-of-care.

At the mutational level, TNBC is primarily a DNA copy-number driven disease[1], harboring a multitude of copy-number alterations (CNAs) containing various driver genes[2]. TNBCs are furthermore characterized by mutations in the TP53 tumor suppressor gene, which occur in more than 80% of cases. Moreover, approximately 50% of TNBCs show loss of BRCA1 or BRCA2, either due to germline or somatic mutations or because of promoter hypermethylation[2]. BRCA1 and BRCA2 are crucial for error-free repair of DNA double-strand breaks via homologous recombination, and loss of these genes results in high levels of chromosomal instability and a specific mutator phenotype. This results in recurrent patterns of CNAs in BRCA-deficient tumors, suggesting that these aberrations contain specific driver genes required for tumorigenesis.

Unfortunately, the high degree of genomic instability in BRCA-deficient TNBCs results in large numbers of CNAs harboring tens-to-thousands of genes, which complicates the identification of putative cancer drivers. To address this issue, several computational approaches have been developed to identify minimal regions that are recurrently gained or lost across tumors[3–6]. Other approaches have complemented these tools with comparative oncogenomic strategies, in which combined analyses of human and mouse tumors are used to identify candidate driver genes that are frequently altered in tumors from both species[7–9]. We have previously used comparative oncogenomics analyses to identify driver genes that were frequently aberrantly amplified or deleted in both mouse and human BRCA1-deficient TNBCs, including the proto-oncogene MYC and the tumor suppressor RB1[10]. However, it is currently still unclear how exactly these putative drivers of BRCA1-deficient TNBC contribute to tumorigenesis, and specifically how they may influence the mutational landscape of the resulting tumors. To address these questions, we generate additional mouse models of BRCA1-deficient TNBC harboring different candidate genes. To overcome the time-consuming nature of generating these mouse models via germline engineering, we develop somatic mouse models of BRCA1-deficient TNBC and we show that these models accurately reflect their germline counterparts. We analyze the resulting tumors to assess the contribution of candidate drivers to BRCA1-associated mammary tumorigenesis and to determine their effect on the copy-number landscape. Finally, by applying comparative oncogenomics to a combined set of germline and somatic BRCA1-deficient TNBCs with MYC overexpression, we identify MCL1 as a key driver and a therapeutic target in these tumors.

## Results

### Driver landscape in human BRCA1-deficient breast cancer.
To determine the mutational landscape of human BRCA1-mutated breast cancer, we performed a meta-analysis by combining datasets from four large-scale breast cancer sequencing studies and extracting the mutational data of all BRCA1-mutated tumors. This analysis identified a total of 80 breast cancers (~1.5%) with a homozygous deletion or an inactivating (putative) driver mutation in BRCA1 (Fig. 1a, Supplementary Table 1). For 18 of these cases (~23%) triple-negative (TN) status could not be determined due to missing or inconclusive immunohistochemistry data. Of

the remaining 62 cases, 40 (~65%) were scored as TNBC. Association with TN status was stronger in tumors from BRCA1 germline mutations carriers (27/30) than in tumors with BRCA1 somatic mutations (13/32).

We next analyzed the mutational landscape of the 80 BRCA1-deficient breast cancer cases, focusing on deleterious mutations, amplifications and homozygous deletions. At the mutational level, these tumors were mainly characterized by mutations in TP53 (52/80, ~65%) and PIK3CA (23/80, ~29%). At the copy-number level, the most prominent events included amplifications of MYC (35/80, ~44%) and several co-amplified genes (e.g., RAD21, EXT1, RECQL4, RSPO2, EPPK1, PLEC) in the same locus (30–34%). MYC is a particularly well-known transcription factor that lies at the crossroad of several growth-promoting pathways and regulates global gene expression, resulting in increased proliferation and influencing many other cellular processes (reviewed in the refs. [11,12]). The MYC oncogene resides in the 8q24 genomic locus, which is among the most frequently amplified regions in breast cancer[13], particularly in TNBC[14]. MYC expression and MYC signaling are aberrantly elevated in TNBC[15,16] and a MYC transcriptional gene signature has been correlated with basal-like breast cancer (BLBC), a subtype typical for human BRCA1-deficient breast cancer[17–19]. Altogether, this confirms that human BRCA1-deficient breast cancers are enriched for TNBCs and are mainly characterized by inactivating mutations in TP53 and amplification of MYC.

### MYC is a potent driver in BRCA1-associated tumorigenesis.
To study the contribution of MYC overexpression to BRCA1-associated mammary tumorigenesis, we initially employed the K14Cre;Brca1$^{F/F}$;Trp53$^{F/F}$ (KB1P) mouse model[20], in which epithelium-specific loss of BRCA1 and p53 leads to the formation of mammary tumors and, to a lesser extent, other epithelial tumors including skin tumors. We used our previously established GEMM-ESC pipeline[21] to generate K14Cre;Brca1$^{F/F}$;Trp53$^{F/F}$;Col1a1$^{invCAG-Myc-IRES-Luc/+}$ (KB1P-Myc) mice with epithelium-specific loss of BRCA1 and p53 and overexpression of MYC. Unfortunately, these mice were more prone to developing non-mammary tumors than KB1P mice and had to be sacrificed around 110 days for skin cancers and thymomas due to expression of K14Cre in these tissues.

To avoid unwanted development of non-mammary tumors, we took a two-pronged approach (Fig. 1b). On one hand, we developed a novel GEMM (WapCre;Brca1$^{F/F}$;Trp53$^{F/F}$, WB1P) in which mammary-specific expression of Cre is driven by the whey acidic protein (Wap) gene promoter. In this WB1P model, female mice spontaneously developed mammary tumors with a median latency of 198 days ($n = 35$, Fig. 1c), which is comparable to the latency of KB1P females (median latency of 197 days, $n = 41$). Similar to KB1P mammary tumors, WB1P tumors were either pure carcinomas (83%) or carcinosarcomas (17%). All tumors were poorly differentiated, negative for ER and PR (Fig. 1d) and showed recombination of the Brca1$^F$ and Trp53$^F$ alleles. On the other hand, we employed a somatic strategy and performed intraductal injection of lentiviral vectors[22–24] expressing the Cre-recombinase (Lenti-Cre) in Brca1$^{F/F}$;Trp53$^{F/F}$ (B1P) females. Tumors from B1P mice injected with Lenti-Cre had a median latency of 238 days after injection ($n = 7$, Fig. 1e), and in terms of their morphology, they were indistinguishable from WB1P tumors (Fig. 1d).

To determine if tumors from these two new mouse models reflected the basal-like subtype typical for human BRCA1-deficient breast cancer, we performed RNA-sequencing on 22 WB1P tumors and 7 tumors from B1P mice injected with Lenti-Cre, and compared their expression profile to tumors from the

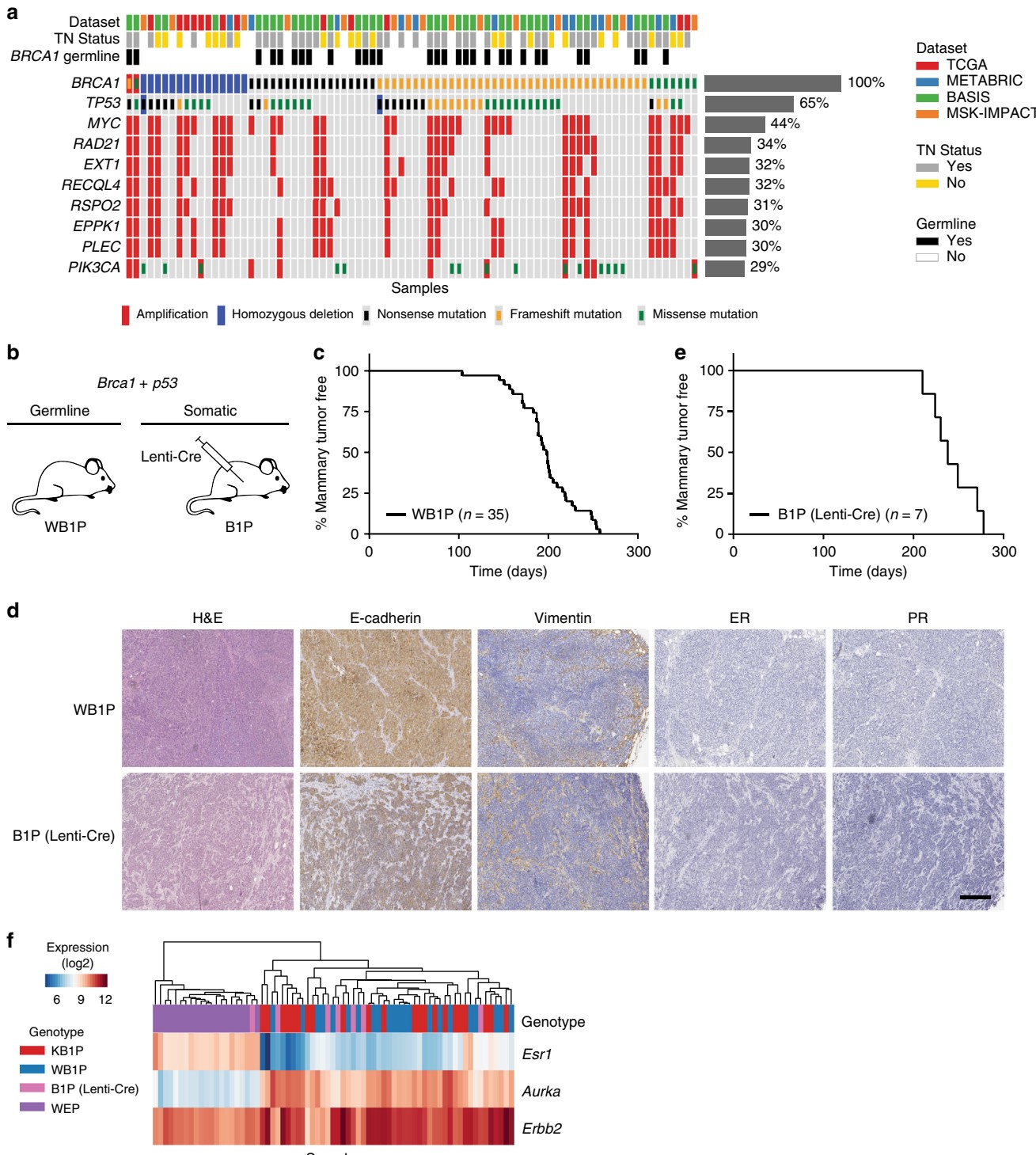

**Fig. 1** Mutational landscape of human *BRCA1*-mutated TNBC and characterization of the WB1P model. **a** Overview of the most common deleterious mutations and copy-number events in 80 *BRCA1*-mutated human breast tumor samples from four large-scale tumor-sequencing studies. **b** Overview of the germline and somatic mouse models for mammary gland-specific inactivation of conditional *Brca1* and *Trp53* alleles. **c** Kaplan–Meier curve showing mammary tumor-specific survival for *WapCre;Brca1^{F/F};Trp53^{F/F}* (WB1P) female mice. **d** Representative hematoxylin and eosin (HE) staining and immunohistochemical detection of E-cadherin, vimentin, ER and PR in WB1P tumors and in tumors from Lenti-Cre injected *Brca1^{F/F};Trp53^{F/F}* (B1P) mice. Bar, 400 μm. **e** Kaplan–Meier curve showing mammary tumor-specific survival of B1P females injected with Lenti-Cre. **f** Unsupervised clustering (Euclidean distance, average linkage) of the WB1P tumors with tumors derived from published mouse models of luminal (*WapCre;Cdh1^{F/F};Pten^{F/F}*, WEP; ref. [25]) and basal-like (*K14Cre;Brca1^{F/F};Trp53^{F/F}*, KB1P; ref. [20]) breast cancer, using a three-genes signature that distinguishes the PAM50 subtypes[26]

KB1P mouse model and a mouse model of luminal breast cancer ($WapCre;Cdh1^{F/F};Pten^{F/F}$, WEP; [25]), using a three-gene signature that distinguishes the PAM50 subtypes[26]. This analysis showed that all $Brca1^{\Delta/\Delta};Trp53^{\Delta/\Delta}$ mouse mammary tumors from the three different mouse models cluster together and are characterized by low expression of $Esr1$ and high expression of the proliferation marker $Aurka$ (Fig. 1f), reflecting the expression profile of human BLBC (Supplementary Figure 1A).

To study the effects of $Myc$ amplification in WB1P mice, we applied the GEMM-ESC strategy[21] to insert the conditional $invCAG-Myc-IRES-Luc$ cassette into the $Col1a1$ locus of WB1P embryonic stem cells (ESC). In the resulting $WapCre;Brca1^{F/F};Trp53^{F/F};Col1a1^{invCAG-Myc-IRES-Luc}/+$ (WB1P-Myc) model, mammary-specific expression of Cre induces inactivation of BRCA1 and p53 and concomitant overexpression of the MYC oncogene accompanied by luciferase expression (Fig. 2a). WB1P-Myc female mice developed multifocal mammary tumors with a median latency of 97 days ($n = 35$, Fig. 2b). These tumors grew exponentially (Supplementary Figure 2A) and animals had to be sacrificed 2–3 weeks after detection of palpable tumors. In contrast to the KB1P-Myc mice, WB1P-Myc mice developed only mammary tumors.

To test if somatic engineering could be used to overexpress MYC in the mammary gland, we performed intraductal injections of Lenti-Cre in $Brca1^{F/F};Trp53^{F/F};Col1a1^{invCAG-Myc-IRES-Luc}/+$ (B1P-Myc, $n = 16$) females (Fig. 2a). Moreover, we also injected lentiviral vectors encoding both Cre and $Myc$ (Lenti-MycP2ACre, Supplementary Figure 3A) in B1P females ($n = 13$) and lentiviral vectors encoding $Myc$ (Lenti-Myc) in WB1P mice ($n = 15$). Mice from all three groups developed mammary tumors with 100% penetrance and specifically in the injected glands (Fig. 2c). B1P-Myc mice injected with Lenti-Cre developed tumors much faster than B1P mice injected with Lenti-Cre (126 days after injection vs. 238 days after injection). B1P females injected with Lenti-MycP2ACre and WB1P females injected with Lenti-Myc developed tumors even faster (median latency of 92 and 61 days after injection, respectively), most likely due to higher $Myc$ expression from the viral constructs than from the knock-in allele (Supplementary Figure 3B).

Histopathological analysis showed that, unlike the WB1P mouse model, WB1P-Myc females developed multifocal tumors that were all carcinomas. However, similar to WB1P tumors, WB1P-Myc tumors were poorly differentiated and ER-/PR-negative (Fig. 2d). Furthermore, they displayed recombined $Brca1$ and $Trp53$ alleles and were sensitive to cisplatin and PARP inhibitors upon transplantation into nude mice (Supplementary Figure 2B). $WapCre;Brca1^{F/+};Trp53^{F/F};Col1a1^{invCAG-Myc-IRES-Luc}/+$ females that were heterozygous for $Brca1^F$ alleles ($n = 20$) developed tumors slightly but significantly slower than $WapCre;Brca1^{F/F};Trp53^{F/F};Col1a1^{invCAG-Myc-IRES-Luc}/+$ mice with homozygous $Brca1^F$ alleles (Supplementary Figure 2C). Histopathologic analysis showed that mammary tumors from the somatic models were indistinguishable from the cognate tumors from the germline models (Fig. 2d). WB1P-Myc tumors showed similar expression levels of $Esr1$ and $Aurka$ as the WB1P tumors, indicating that they retained their basal-like subtype (Supplementary Figure 2D). Besides this, WB1P-Myc tumors showed high mRNA and protein levels of MYC compared to WB1P tumors, demonstrating successful expression of the knock-in allele (Supplementary Figure 2E-F). Unsupervised clustering of RNA-seq data from tumors from the somatic models confirmed that they also retained their basal-like phenotypes, and PCA analysis showed that these tumors also resemble their counterparts from the germline models in terms of their global gene expression profiles (Supplementary Figure 3C-E). Taken together, these data provide functional

validation in germline and somatic models of the role of MYC in BRCA1-associated mammary tumorigenesis.

**Loss of PTEN and RB1 collaborates with MYC in tumorigenesis.** After MYC amplification, the next most common alterations in our analysis of the human BRCA1-deficient TNBCs were mutations and/or amplifications of $PIK3CA$ (23/80 cases), indicating that activation of PI3K signaling is an important driver in this breast cancer subtype (Fig. 1a). Indeed, in addition to $PIK3CA$ mutation/amplification, heterozygous or homozygous loss of $PTEN$ (a negative regulator of PI3K signaling) was observed in 29/80 and 6/80 cases, respectively (Supplementary Table 1). Genetic alterations of $PIK3CA/PTEN$ and $MYC$ co-occurred in ~29% of all tumors analyzed (23/80 cases), indicating that MYC overexpression and PI3K pathway activation collaborate in BRCA1-related breast tumorigenesis.

To assess if activation of PI3K signaling via loss of PTEN collaborates with MYC overexpression in BRCA1-deficient TNBC, we developed $WapCre;Brca1^{F/F};Trp53^{F/F};Col1a1^{invCAG-Cas9-IRES-Luc}/+$ (WB1P-Cas9) mice with mammary-specific loss of BRCA1 and p53 and concomitant expression of Cas9. We then cloned and validated lentiviral vectors encoding a nontargeting sgRNA (sgNT) or a sgRNA targeting the seventh exon of $Pten$ (sg$Pten$), in combination with a $Myc$-overexpression cassette. Since also RB1 loss has been implicated in BRCA1-deficient breast cancer[27] and MYC-driven TNBC[28], we also generated a similar lentiviral vector encoding MYC and a sgRNA targeting the second exon of $Rb1$ (sg$Rb1$). These lentiviral vectors (Lenti-sgNT-Myc, Lenti-sg$Pten$-Myc, and Lenti-sg$Rb1$-Myc) were injected intraductally into WB1P-Cas9 females (Fig. 2e) resulting in tumor formation with high penetrance and very short latency (70, 30, and 52 days after injection, respectively; $n = 14$, 12, and 14, respectively, Fig. 2f). Genomic DNA of mammary tumors from Lenti-sg$Pten$-Myc and Lenti-sg$Rb1$-Myc injected WB1P-Cas9 mice showed extensive modification of the target gene (Fig. 2g; Supplementary Figure 4A-B), with a strong bias towards indels resulting in frameshift mutations, supporting homozygous inactivation of the tumor suppressor genes. Together, these results demonstrate that activation of PI3K signaling and RB1 loss collaborate with MYC in BRCA1-deficient TNBC.

**MYC overexpression reshapes the copy-number landscape.** To identify additional collaborating driver genes in BRCA1-deficient TNBC, we decided to characterize the CNA landscape of WB1P and WB1P-Myc tumors, with the assumption that recurrent CNAs in these tumors might underscore a conserved selective pressure towards the specific gain or loss of cancer genes that collaborate with loss of BRCA1 and p53—alone or in combination with MYC overexpression—during TNBC development. We therefore performed DNA copy-number sequencing (CNV-seq) on 39 WB1P tumors and identified recurrent CNAs using RUBIC[6]. This analysis showed that WB1P tumors exhibit a high degree of genomic instability and harbor a multitude of recurrent gains and losses (Fig. 3a; Supplementary Figure 5A). The most evident of these events was a focal amplification on chromosome 6 containing the $Met$ oncogene. Besides $Met$, we also identified a recurrent loss on chromosome 14 (harboring $Rb1$) and several amplifications on chromosome 15 (containing $Myc$), in line with our previous studies in KB1P mice[10].

Remarkably, CNV-seq of 19 WB1P-Myc tumors showed a dramatically reshaped copy-number landscape (Fig. 3b), with significantly fewer CNAs compared to the WB1P model (Fig. 3c; $P < 0.00001$, two-sided Mann–Whitney $U$-test). To determine if the decreased number of CNAs observed in WB1P-Myc tumors was not simply a result of the shortened tumor latency, we

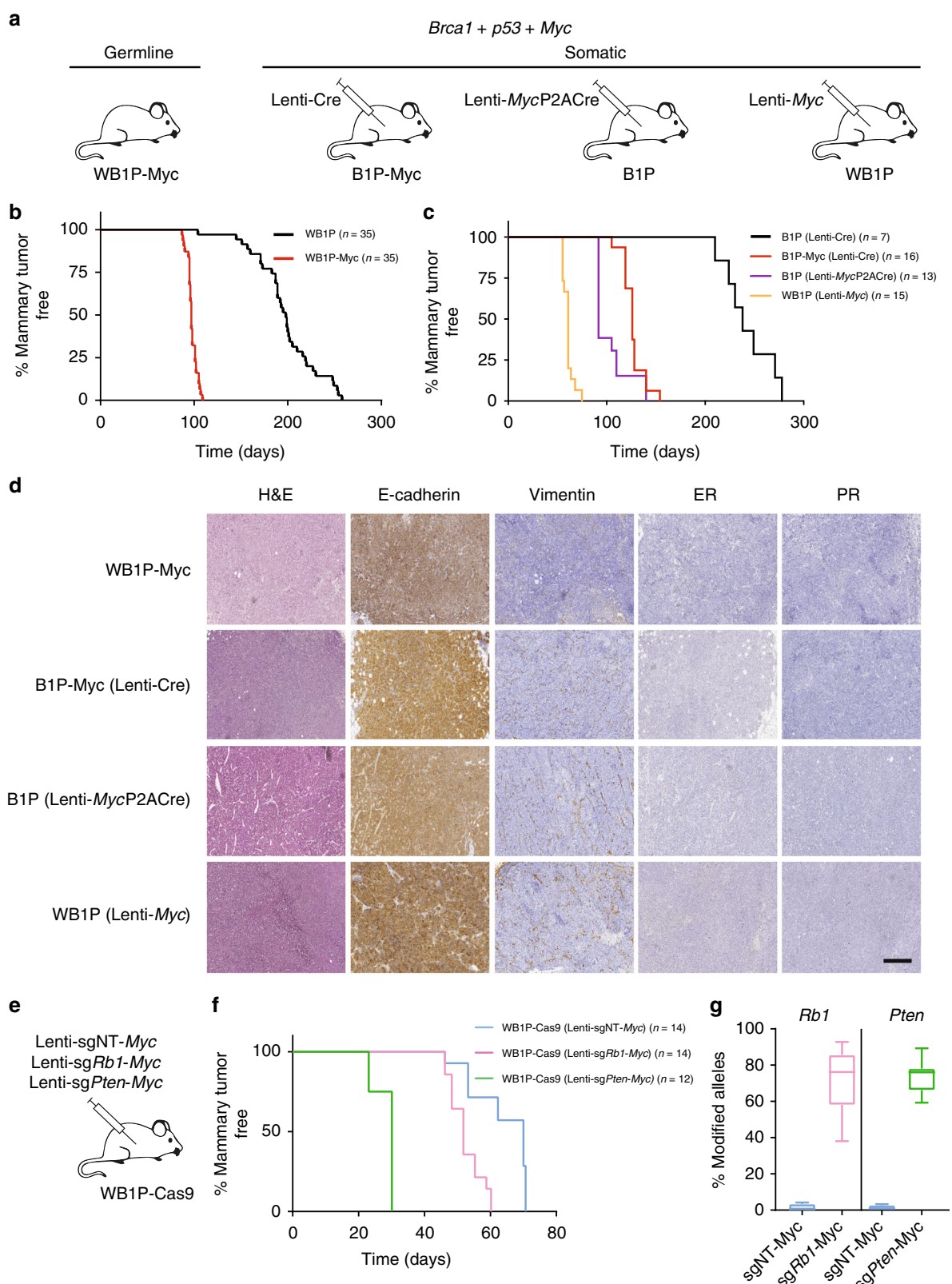

generated $WapCre;Brca1^{F/F};Trp53^{F/F};Col1a1^{invCAG-Met-IRES-Luc}/+$ (WB1P-Met) mice containing the $Met$ oncogene, which we found frequently amplified in the WB1P tumors. Similar to WB1P-Myc females, WB1P-Met female mice developed multifocal mammary tumors with a short latency of 89 days ($n = 11$, Supplementary Figure 6A). All WB1P-Met tumors were classified as poorly

differentiated ER/PR-negative ductal carcinomas and showed MET overexpression and active MET signaling (Supplementary Figure 6B-E). These data confirm the previously reported role of MET in the onset and progression of TNBC[29]. CNV-sequencing of WB1P-Met tumors ($n = 20$) showed an intermediate number of CNAs (Supplementary Figure 6F), which was lower than the

**Fig. 2** Validation of additional drivers in WB1P mice using germline and somatic engineering. **a** Overview of the germline and somatic mouse models for mammary gland-specific *Myc* overexpression in mice with conditional *Brca1* and *Trp53* alleles. **b** Kaplan–Meier curves showing mammary tumor-specific survival for the different genotypes. *WapCre;Brca1^F/F^;Trp53^F/F^;Col1a1^invCAG-Myc-IRES-Luc^/+* (WB1P-Myc) females showed a reduced mammary tumor-specific survival compared to WB1P littermates (97 days vs. 198 days; ****$P < 0.0001$ by Mantel-Cox test). **c** Kaplan–Meier curves showing mammary tumor-specific survival for the different non-germline models. *Brca1^F/F^;Trp53^F/F^;Col1a1^invCAG-Myc-IRES-Luc^/+* (B1P-Myc) females injected with Lenti-Cre, B1P females injected with Lenti-MycP2ACre and WB1P females injected with Lenti-Myc showed a reduced mammary tumor-specific survival compared to B1P female mice injected with Lenti-Cre (respectively 126, 92, and 61 days after injection vs. 238 days after injection; ****$P < 0.0001$ by Mantel-Cox test).
**d** Representative hematoxylin and eosin (HE) staining and immunohistochemical detection of E-cadherin, vimentin, ER and PR in tumors from WB1P-Myc females and in tumors from Lenti-Cre injected B1P-Myc mice, Lenti-MycP2ACre injected B1P mice and Lenti-Myc injected WB1P mice. Bar, 400 μm.
**e** Overview of the intraductal injections performed in *WapCre;Brca1^F/F^;Trp53^F/F^;Col1a1^invCAG-Cas9-IRES-Luc^/+* (WB1P-Cas9) females with high-titer lentiviruses encoding *Myc* and either a non-targeting (NT) sgRNA (Lenti-sgNT-Myc), a sgRNA targeting exon 2 of *Rb1* (Lenti-sgRb1-Myc) or a sgRNA targeting exon 7 of *Pten*. **f** Kaplan–Meier curves showing mammary tumor-specific survival for the different models. WB1P-Cas9 females injected with Lenti-sgPten-Myc and *Lenti-sgRb1*-Myc showed a reduced mammary tumor-specific survival compared to WB1P-Cas9 female mice injected with Lenti-sgNT-Myc (respectively 30 and 52 days after injection vs. 70 days after injection, ****$P < 0.0001$ and ***$P < 0.001$ by Mantel-Cox test). **g** Boxplots depicting the fraction of modified *Rb1* and *Pten* alleles in tumors from WB1P-Cas9 mice injected with Lenti-sgNT-Myc, Lenti-sgRb1-Myc and Lenti-sgPten-Myc. Boxes extend from the third (Q3) to the first (Q1) quartile (interquartile range, IQR), with the line at the median; whiskers extend to Q3 + 1.5 × IQR and to Q1 − 1.5 × IQR

---

WB1P tumors but significantly higher than the WB1P-Myc tumors ($P < 0.001$, one-sided Mann–Whitney $U$-test). This demonstrates that the observed differences in CNA load are not merely a function of tumor latency but also of the driver gene. Moreover, the validation of MET as a potent driver in BRCA1-associated tumorigenesis underscores the potential of iterative analysis of CNAs in progressively complex mouse models as an approach for identifying putative cancer genes that promote tumorigenesis in specific genetic contexts.

**Comparative oncogenomics identifies MCL1 as a driver gene**. Our RUBIC analyses showed that most of the CNAs identified in WB1P tumors were no longer present in WB1P-Myc tumors, suggesting an increased evolutionary pressure to acquire only specific driver mutations (Fig. 3b). Interestingly, a small number of losses were retained, including the *Rb1*-associated loss on chromosome 14, further supporting *Rb1* as a collaborating driver in MYC-driven BRCA1-deficient mammary tumors. Focusing on novel events, we identified a strongly recurrent amplicon on chromosome 11 encompassing the *Col1a1* locus in which we introduced the *invCAG-Myc-IRES-Luc* cassette. The recurrent amplification of this locus suggests that WB1P-Myc tumors underwent a selection for increased MYC expression via amplification of the conditional *Myc* knock-in allele. Besides this, we also identified novel recurrent amplifications on chromosome 3 and chromosome 15, which were syntenic with human 1q and 22q loci, respectively, which are commonly amplified in breast cancer patients.

To identify additional driver genes in MYC-driven BRCA1-deficient TNBC, we used a comparative oncogenomics strategy to select candidate genes that are recurrently aberrated in both WB1P-Myc tumors and human BLBCs from TCGA. In this approach (outlined in Fig. 3d), we first identified candidate drivers in both species individually using RUBIC. For the mouse tumors, we combined CNV-seq data of tumors from both the WB1P-Myc GEMM and the somatically engineered MYC-driven B1P models to increase our sample size, based on the observation that these tumors share the same distinctive CNA profile (Supplementary Figure 5B-C). Next, we mapped genes between species using mouse-human orthologs and took the intersection of both candidate lists. Finally, to prioritize genes that show differences in expression, we filtered the remaining candidates for genes with a positive Spearman correlation (>0.2) between their expression and copy-number status.

After applying this strategy, we focused on genes residing in the recurrent amplifications on mouse chromosomes 3 and 15, as these were the most striking new events in the WB1P-Myc model.

The recurrent amplification on chromosome 11 containing the conditional *Myc* knock-in allele in the *Col1a1* locus was excluded from this analysis. While this did not identify any candidate genes in the peak on chromosome 15 (mainly due to a lack of orthologous, recurrently aberrated genes), it did identify a list of 12 candidate genes residing in the peaks on mouse chromosome 3 (Fig. 3e) and human chromosome 1q (Fig. 3f). To identify potential drivers in this list of candidates, we derived organoids from a WB1P-Myc mammary tumor using our recently established methodology[30]. We next performed a fitness screen in these WB1P-Myc organoids with a focused lentiviral shRNA library targeting candidate genes. This screen showed a marked depletion for shRNAs targeting *Mcl1* (Fig. 4a), indicating that MCL1 expression is essential for growth of WB1P-Myc tumor cells. In line with this, WB1P-Myc tumors showed strongly elevated expression of MCL1 compared to WB1P tumors (Fig. 4b).

To determine whether MCL1 cooperates with MYC in driving BRCA1-deficient TNBC, we generated a lentiviral vector encoding both *Cre* and *Mcl1* (Lenti-Mcl1P2ACre, Supplementary Figure 7A) and injected this lentivirus intraductally into B1P and B1P-Myc females ($n = 7$ and $n = 11$, respectively) to achieve simultaneous Cre-mediated recombination of the conditional alleles and overexpression of *Mcl1* (Fig. 4c). Co-expression of MCL1 and Cre in B1P and B1P-Myc mice resulted in a significant decrease in tumor latency compared to mice in which only Cre was delivered (180 vs. 238 days and 70 vs. 126 days, respectively; Fig. 4d; Supplementary Figure 7B). MCL1 overexpression appeared to relieve pressure for chromosome 3 amplification in the resulting tumors (Supplementary Figure 7C-D). Conversely, MCL1 silencing in WB1P-Myc organoids resulted in *Myc* downregulation (Supplementary Figure 7E). Altogether, these analyses identify *Mcl1* as an important driver gene in the recurrent amplification on mouse chromosome 3 and demonstrate that MCL1 effectively collaborates with MYC in BRCA1-associated breast tumorigenesis.

**MCL1-inhibition synergizes with PARP-inhibition in PDXs**. Our focused shRNA library screen showed that knockdown of *Mcl1* is detrimental to WB1P-Myc organoids. To test whether WB1P-Myc tumors are also sensitive to pharmacological MCL1-inhibition, we tested the in vitro sensitivity of WB1P and WB1P-Myc organoids to the selective MCL1 inhibitor S63845[31], which was recently shown to display activity in patient-derived xenograft (PDX) models of breast cancer[32]. In contrast to other BH3 mimetics, S63845 binds to human MCL1 at sub-nanomolar concentrations whereas it does not display detectable binding to

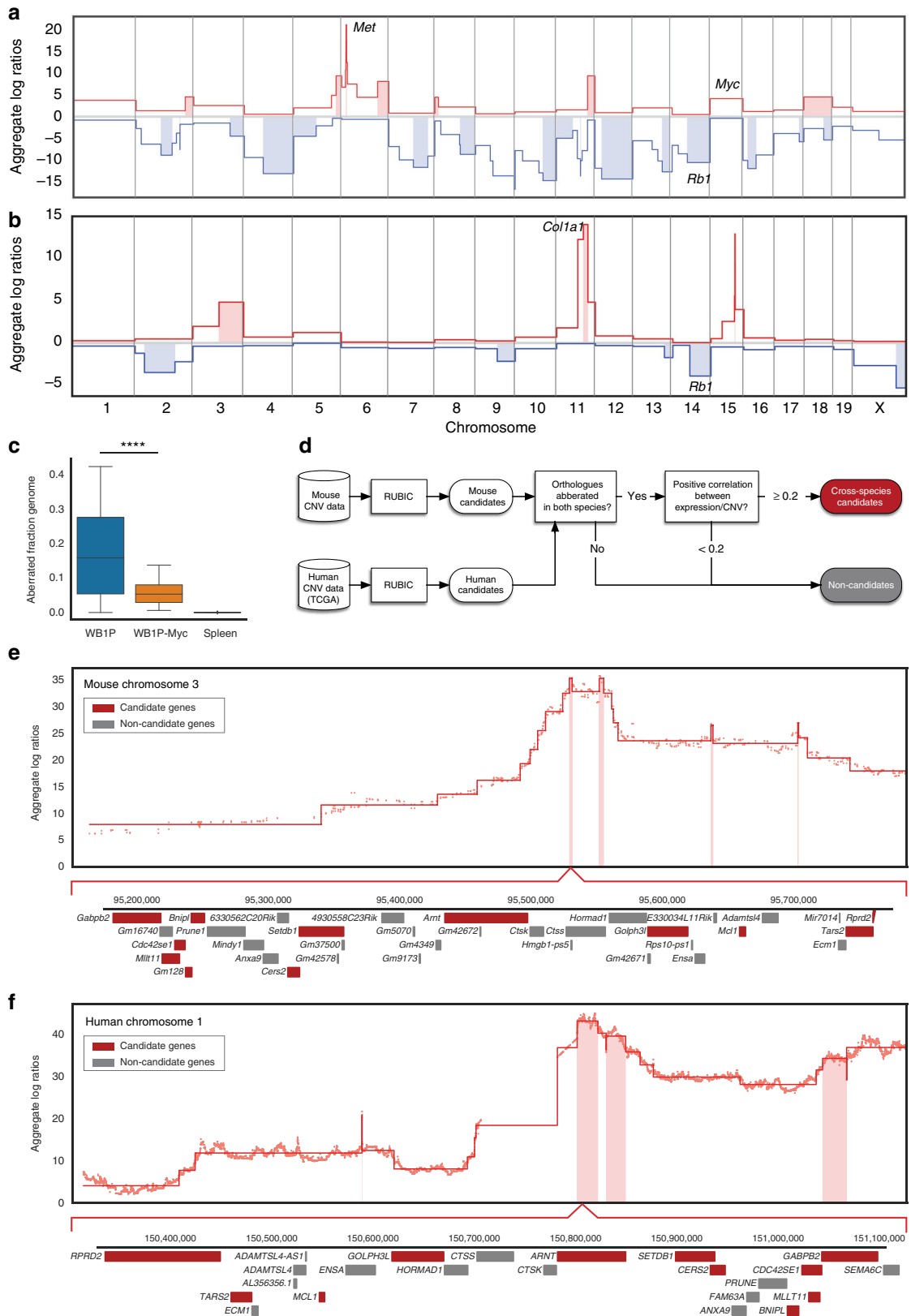

other anti-apoptotic family members[31]. Proliferation assays indicated that WB1P-Myc organoids were more sensitive to S63845 than WB1P organoids (Fig. 4e). To examine the response of WB1P and WB1P-Myc tumors to MCL1-inhibition, we transplanted WB1P and WB1P-Myc organoids orthotopically into nude mice ($n = 10$ per line) and tested the response of the tumor outgrowths to S63845. However, in this setting we did not observe a differential sensitivity to MCL1-inhibition, as none of the tumors responded to S63845 at the tested dose (Fig. 4f).

**Fig. 3** Identification of candidate drivers in WB1P-Myc tumors using comparative oncogenomics. **a**, **b** Genome-wide RUBIC analysis of CNV profiles of WB1P tumors (**a**) and WB1P-Myc tumors (**b**). Significant amplifications and deletions are marked by light red and blue columns, respectively. **c** Genomic instability of WB1P and WB1P-Myc tumors. Scores for spleen samples from WB1P mice are shown as reference; ****$P < 0.0001$ (two-sided Mann–Whitney $U$-test). Boxes extend from the third (Q3) to the first (Q1) quartile (interquartile range, IQR), with the line at the median; whiskers extend to Q3 + 1.5 × IQR and to Q1 − 1.5 × IQR. See Materials and Methods for more details. **d** Flowchart illustrating the comparative oncogenomics analysis pipeline used for the identification of additional cancer driver genes. **e** Chromosome 3 RUBIC analysis of the combined CNV profiles of the tumors from germline and somatic mouse models overexpressing *Myc* in the mammary gland. Significant amplifications are marked by light red columns. Genes residing in the minimal amplicon of chromosome 3 are shown. Cross-species candidate genes surviving filter criteria are colored in red. **f** Chromosome 1 RUBIC analysis of the CNV profiles of human TNBC. Significant amplifications are marked by light red columns. Orthologs of the genes shown in **e** are shown. Cross-species candidate genes surviving filter criteria are colored in red

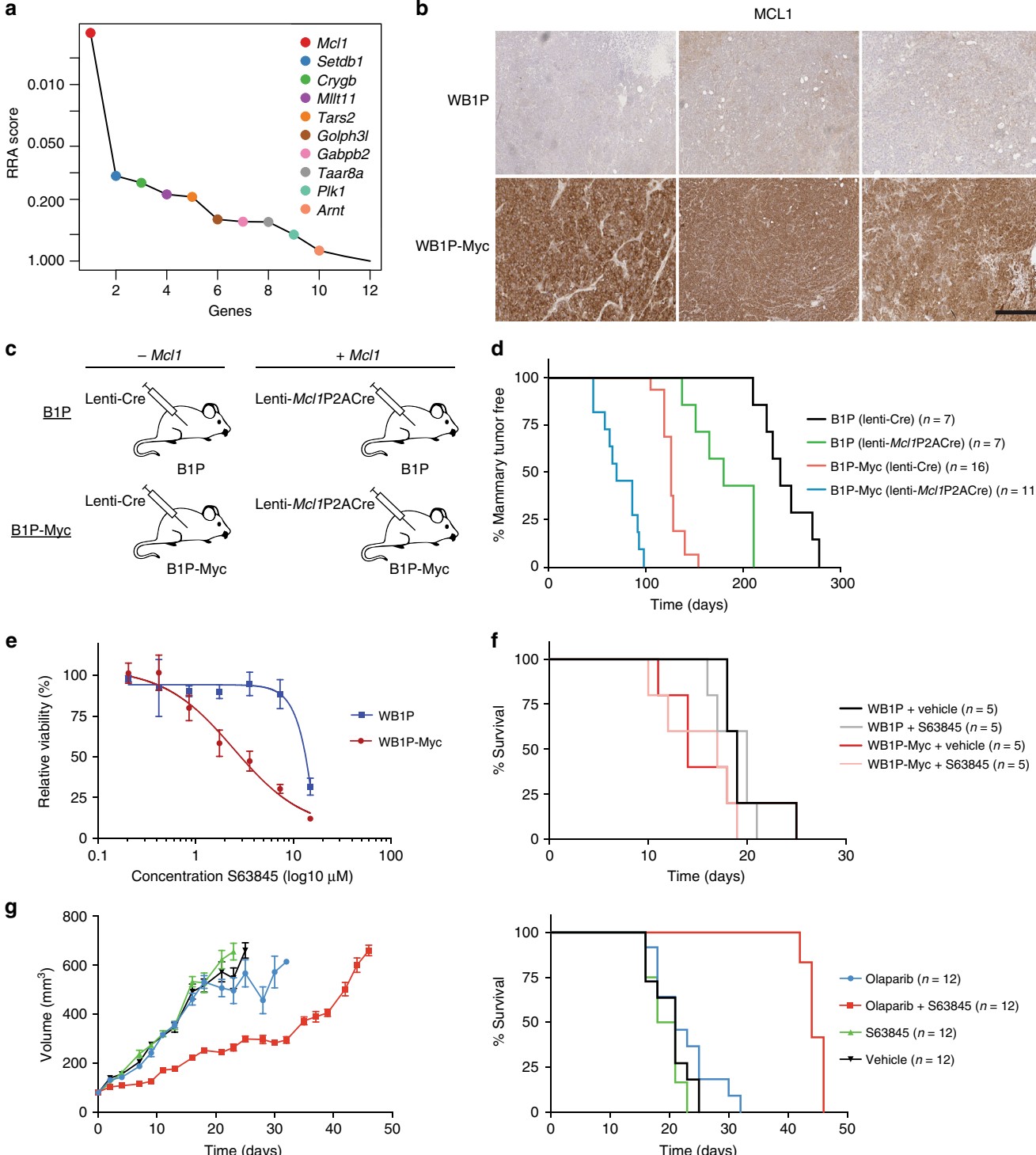

**Fig. 4** Validation of MCL1 as a druggable driver in *BRCA1*-mutated TNBC. **a** MAGeCK software was used to compute RRA scores for all genes included in our focused shRNA library, showing depletion of *Mcl1* shRNAs in WB1P-Myc organoids. **b** Immunohistochemical detection of MCL1 in multiple independent WB1P and WB1P-Myc tumors. Bar, 400 μm. **c** Overview of the non-germline mouse models for mammary-specific *Mcl1* overexpression. **d** Kaplan–Meier curves showing mammary tumor-specific survival for the different models. B1P and B1P-Myc females injected with Lenti-Mcl1P2ACre showed a reduced mammary tumor-specific survival compared to B1P and B1P-Myc female mice injected with Lenti-Cre, respectively (180 days after injection vs 238 days after injection; **P < 0.01 by Mantel-Cox test; 70 days after injection vs 126 days after injection; ****P < 0.0001 by Mantel-Cox test). **e** In vitro response of WB1P and WB1P-Myc organoids to MCL1 inhibitor S63845. Error bars represent standard error of the mean. Experiment was performed in triplicate. **f** In vivo response of organoid-derived WB1P and WB1P-Myc tumors to S63845, as visualized by Kaplan–Meier curves. WB1P and WB1P-Myc organoid lines were transplanted in the fourth mammary fat pad of nude mice. When tumors had reached a size of 100 mm$^3$, mice were treated with 25 mg kg$^{-1}$ S63845 (i.v. once-weekly for 5 weeks) or vehicle. **g** Response of the *BRCA1*-mutated TNBC PDX-110 xenograft model to S63845 and the PARP inhibitor olaparib, as visualized by tumor volume curves (left) and Kaplan–Meier curves (right). Single-cell suspensions of PDX-110 were transplanted in the fourth mammary fat pad of *NOD-SCID-IL2Rγ$_c$$^{-/-}$* mice. When tumors had reached a size of 100 mm$^3$, mice were treated with 25 mg kg$^{-1}$ S63845 (i.v. once-weekly for 4 weeks), 50 mg kg$^{-1}$ olaparib (i.p. 5 days out of 7 for 4 weeks), both drugs or vehicle. Combination therapy with S63845 and olaparib prolonged survival compared to olaparib monotherapy (****P < 0.0001 by Mantel-Cox test). Error bars represent standard error of the mean

Based on the anti-apoptotic role of MCL1, we reasoned that MCL1-inhibition might be most effective when combined with a pro-apoptotic drug, as previously demonstrated in PDX models of HER2-amplified breast cancer and TNBC treated with trastuzumab and docetaxel, respectively[32]. We therefore investigated whether S63845 could enhance the efficacy of the clinical PARP inhibitor (PARPi) olaparib, which is currently used for treatment of *BRCA1*-mutated breast cancer patients. As the olaparib-sensitivity of WB1P and WB1P-Myc tumors was too high to assess any synergistic effect of MCL1-inhibition (Supplementary Figure 2B; Supplementary Figure 8A-B), we turned to a PDX model of *BRCA1*-mutated TNBC (PDX-110), which expresses relatively high levels of MYC and MCL1 and shows limited sensitivity to PARPi[32]. To assess the effect of combined inhibition of PARP and MCL1 in this model, PDX tumor cells were orthotopically injected into *NOD-SCID-IL2Rγ$_c$$^{-/-}$* mice ($n = 48$), which were randomly allocated to vehicle-, single- or double-treatment arms once tumors reached a volume of 100 mm$^3$. Remarkably, while treatment with S63845 or olaparib alone did not elicit a clinical response, tumor growth was considerably inhibited upon treatment with both drugs and tumors relapsed only when treatment was stopped after 4 weeks (Fig. 4g).

## Discussion

In this work, we have used both germline and somatic engineering approaches to rapidly test candidate cancer drivers in the WB1P mouse model of BRCA1-deficient TNBC. Using this approach, we validated MYC, MET, PTEN, and RB1 as bona fide drivers of BRCA1-associated tumorigenesis and showed that MYC overexpression dramatically changes the mutational landscape of the resulting tumors. Finally, by applying a comparative oncogenomics strategy to uncover additional culprits of tumorigenesis, we identified MCL1 as a druggable cancer driver that collaborates with MYC in BRCA1-deficient TNBC.

An important challenge of modeling cancer in mice, is that it requires technology that allows rapid introduction of new driver mutations to quickly create a variety of compound-mutant mouse models containing different combinations of candidate genes. Such experiments are difficult to perform using germline engineering approaches, which generally involve extensive cross-breeding of single-mutant mice to produce animals carrying the desired combination of mutations. Here, we have shown that somatic engineering using lentiviral vectors for overexpression of cDNAs and CRISPR-mediated in situ gene editing provides an effective alternative for rapid generation of novel mouse models of BRCA1-deficient TNBC. The limitations of cDNA-based overexpression systems—which may not fully recapitulate the desired expression levels of candidate genes—might be alleviated

by implementing novel technologies for CRISPR-mediated transcriptional control (CRISPRi/CRISPRa) and base-editing of endogenous genes. We therefore expect that our somatic engineering methodology can ultimately be used to generate refined breast cancer models containing a wide variety of fine-tuned (epi)genetic permutations.

One of the key advantages of this type of iterative mouse modeling, is that it highlights the profound effect that additional driver genes can have on the mutational landscape of a baseline tumor model. This indicates that the evolutionary pressure that tumor cells experience depends strongly on the combination of driver mutations, which push cells down a specific path to acquire additional aberrations that most effectively collaborate with the pre-existing events. This notion has two important implications. First, it means that it is crucial to study driver genes in the appropriate genetic contexts as observed in human tumors, as they may have very different effects in different backgrounds. Second, predominant changes in the mutational landscape likely indicate that the pre-existing driver(s) push tumors down a relatively restricted evolutionary path, which might be exploited therapeutically.

A clear example is provided by our study, showing that MYC-overexpression pushes the evolution of BRCA1-deficient TNBC towards amplification of a druggable driver, MCL1. Although MYC is a potent inducer of cell proliferation, supraphysiologic overexpression of *MYC* also has pro-apoptotic effects and is generally not tolerated in non-transformed cells[33]. This suggests that tumor cells need to acquire additional alterations in other, collaborating cancer driver genes to counteract MYC-induced apoptosis. MCL1 belongs to the Bcl-2 family and is involved in the inhibition of apoptosis[34]. While it cannot be excluded that MYC overexpression reshapes evolution of BRCA1-deficient TNBCs via negative selection of tumor cell clones with high levels of CNAs, amplifications of *MCL1* might be particularly selected for as they may counteract the pro-apoptotic effects of MYC overexpression. *MCL1* amplifications have been identified in a variety of tumor types, including breast cancer[35], where they correlate with poor survival[36,37]. Although the commonly amplified chromosome 1q region (where *MCL1* resides) might harbor additional driver genes, MCL1 is the main pro-survival protein upregulated in TNBC[38], and amplification of *MCL1* has been implicated in resistance to multiple therapies used in patients with TNBC, where it is often co-amplified with *MYC*[39,40]. We found amplification of *MCL1* in 15% of the *BRCA1*-mutated TNBCs we analyzed (12/80 cases), and two-thirds of these cases showed co-amplification of *MYC* (Supplementary Figure 8C). MCL1 and MYC were also shown to cooperate in mouse models of leukemia and non-small cell lung cancer (NSCLC), and co-expression of these two factors correlates

with poor NSCLC patient survival[41,42]. This suggests that MCL1 inhibition might be particularly effective against *MYC*-over-expressing tumors by exposing them to MYC-induced apoptosis. MCL1 has only recently been recognized as an important therapeutic target, and currently several MCL1 inhibitors are being tested in preclinical trials, where they are showing promising activity, especially in combination therapies[32,43]. Five phase-I clinical trials are currently ongoing for testing MCL1 inhibitors in patients with hematopoietic malignancies (NCT02675452; NCT02992483; NCT02979366; NCT03672695; NCT03465540). Our study demonstrates that MCL1 inhibition considerably enhances response of *BRCA1*-mutated TNBC to the clinical PARPi olaparib and suggests that this combination should be prioritized for clinical evaluation, especially in *BRCA1*-mutated cancer patients with poor response to PARPi monotherapy.

Another example of identification of druggable drivers in mouse models of BRCA1-deficient TNBC was recently provided by Liu et al.[44], who analyzed transcriptional and CNA profiles of mammary tumors from our previously published KB1P and *K14Cre;Trp53*[F/F] (KP) models[20]. This analysis yielded a spectrum of somatic genetic alterations putatively driving tumor evolution, including gene-fusions and chromosomal amplifications and deletions (including recurrent amplification of *Met* and *Myc* and deletion of *Rb1*). Interestingly, even though KB1P and KP tumors were following diverse evolutionary trajectories, most tumors displayed enhanced activation of MAPK and/or PI3K signaling and could be treated with inhibitors specific for the aberrated drug target.

In summary, we applied novel germline and somatic technologies to functionally validate the role of candidate drivers in BRCA1-deficient TNBC in vivo at unprecedented speed. Our integrate approach revealed a profound effect of MYC over-expression on tumor evolution and identified MCL1 as a critical and druggable dependency in BRCA1-deficient TNBC with high expression of MYC. Combined inhibition of MCL1 and PARP might benefit a subset of *BRCA1*-mutated TNBC patients and warrants further investigation.

## Methods

**Lentiviral vectors**. The Lenti-Cre vector (pBOB-CAG-iCRE-SD, Addgene plasmid #12336) was a kind gift of Lorenzo Bombardelli. Lenti-MycP2ACre and Lenti-Mcl1-P2ACre were cloned as follows. AgeI and SalI were used to remove GFP-T2A-puro from the SIN.LV.SF-GFP-T2A-puro[45]. P2ACre was synthesized with AgeI-SalI overhangs and inserted as AgeI-SalI fragment in the SIN.LV.SF-GFP-T2A-puro backbone, resulting in SIN.LV.SF-P2ACre. *Myc* and *Mcl1* murine cDNAs were isolated with BamHI-AgeI overhangs using standard PCR from cDNA clones (Clone 8861953, Source BioScience; Clone 3491015, Dharmacon) and inserted as BamHI-AgeI fragments into the SIN.LV.SF-P2ACre vector. The Lenti-sg*Rb1*-Myc, Lenti-sg*Pten*-Myc and Lenti-sgNT-Myc vectors were cloned as follows. *Myc* cDNA was isolated with XbaI-XhoI overhangs using standard PCR from the Lenti-MycP2ACre vector, and inserted as XbaI-XhoI fragment into pGIN, a lentiviral vector for sgRNA overexpression[46]. The non-targeting sgRNA (TGATTGGGGGTCGTTCGCCA) and sgRNAs targeting mouse *Rb1* exon 2 (TCTTACCAGGATTCCATCCA) and mouse *Pten* exon 7 (CCTCAGCCATTGC CTGTGTG) were cloned as described[47]. All vectors were validated by Sanger sequencing. Concentrated stocks of VSV-G pseudotyped lentivirus were produced by transient co-transfection of four plasmids in 293T[48]. Lentiviral titers were determined using the qPCR lentivirus titration kit from Abm (LV900).

**Cell culture**. 293T cells for lentiviral production (American Tissue Culture Collection, ATCC) and Cre-reporter 293T cells containing a lox-STOP-lox-GFP cassette were cultured in Iscove's medium (Invitrogen Life Technologies) containing 10% FBS, 100 IU ml$^{-1}$ penicillin, and 100 μg ml$^{-1}$ streptomycin. Transductions were performed by adding diluted viral supernatant to the cells in the presence of 8 μg ml$^{-1}$ polybrene (Sigma). Cells were transduced for 24 h, after which medium was refreshed. Harvesting of cells for flow cytometry or immunoblotting was performed 5 days after transduction.

**Flow cytometry**. Cells were collected 5 days after transduction, washed in PBS, fixed in Fixation Buffer (BD Biosciences) and permeabilized with Perm Buffer III (BD Biosciences). They were then stained using the primary rabbit antibody anti-Myc (1:1000, Abcam ab32072) or anti-Mcl1 (1:1000, Cell Signaling 94296S) for 30 min at 4 degrees, washed in PBS and incubated for 15 min with an AlexaFluor647-conjugated secondary anti-rabbit antibody (1:1000, Thermofisher). Stained cells were analyzed using a Becton Dickinson LSR FORTESSA. GFP and AlexaFluor647 expression of viable cells was measured using a 488 nm and 640 nm laser for excitation, respectively. Data analysis was performed using FlowJo software version 7.6.5.

**PCRs and TIDE analyses**. Amplification of *Rb1* exon 2 and *Pten* exon 7 was performed with specific primers spanning the target sites (FW_Rb1: TCACCA TGCTAGCAGCTCTTC; RV_Rb1: AGCCAGTTCAATGGTTGTGGG; FW_Pten: TGTATTTAACCACACAGATCCTCA; RV_Pten: AACAAACTAAGGGTCGG GGC) and 1 μg DNA template using the Q5 high-fidelity PCR kit from NEB. Amplicons were run on 1% agarose gel and gel-purified using the Isolate II PCR and Gel kit (Bioline). PCR products were Sanger-sequenced using the FW primer and CRISPR/Cas9-induced editing efficacy was predicted and quantified as described ([http://tide.nki.nl][49]). Untransduced cells were taken along as a control in each gRNA amplification.

**Immunoblotting**. Protein lysates were made using lysis buffer (20 mM Tris pH 8.0, 300 mM NaCl, 2% NP40, 20% glycerol, 10 mM EDTA) complemented with protease inhibitors (Roche) and quantified using the BCA Protein Assay Kit (Pierce). Protein lysates were loaded onto a 3–8% Tris-Acetate gradient Gel (Invitrogen) and transferred overnight onto PVDF membrane (Millipore, methanol pre-wetted) in transfer buffer (38 mM glycine, 5 mM TRIS and 0.01% SDS in PBS-T (0.5% Tween-20). Membranes were blocked in 5% ELK in TBS-T after which they were stained for four hours at room temperature using the primary antibody anti-Met (1:1000, Cell Signaling 4560S), anti-phosphoMet (1:1000, Cell Signaling 3077S), anti-Myc (1:1000, Abcam ab32072), or anti-Mcl1 (1:1000, Cell Signaling 94296S). Membranes were washed three times with 1% milk in PBS-T and incubated for 1 h with an HRP-conjugated secondary antibody (1:2000, DAKO). Stained membranes were washed three times in 1% milk in PBS-T and developed using SuperSignal ECL (Pierce).

**Organoids culture**. WB1P and WB1P-Myc mammary tumor organoids were isolated and cultured as described[30]. For cell viability assays, organoids were seeded (100,000 cells per well) in 40 μl complete mouse media/BME mixture on 24-well suspension plates and cultured for 5 days in the presence of S63845 (ApEXBiO). Cell viability was assessed using the resazurin-based Cell Titer Blue assay following manufacturer's protocol (Promega). Cell viability experiments were performed three times in triplicate and data were analyzed with GraphPad Prism statistical software using nonlinear regression and extra sum-of-squares *F*-test. For the focused shRNA library screen in WB1P-Myc organoids[30], a small library of shRNA targeting candidate genes was built from the Mission shRNA collection (mouse TRC v1.0 collection) by pooling shRNAs targeting candidate genes (*Mcl1*, *Gabpb2*, *Arnt*, *Setdb1*, *Tars*, *Golph3l*, *Lass2*, and *Mllt11*) and control genes (*Plk1*, *Nlrp5*, *Crygb*, and *Taar8a*). Organoids were transduced at MOI 0.3 and analyzed for shRNA representation at day 0, 7, and 14. MAGeCK software was used to compute RRA scores for all genes to identify relative shRNA depletion.

**Meta-analysis of four human breast cancer datasets**. Curated copy-number and mutation data for the METABRIC, TCGA and MSK-impact datasets were down-loaded from cBioPortal ([http://cbioportal.org], 13/10/2017), after which the downloaded mutation data was filtered for deleterious mutations (Missense_Mutation, Nonsense_mutation, Frame_Shift_Del and Frame_Shift_Ins). Similarly, copy-number data were filtered for high-level amplifications (amp) or homozygous deletions (homdel). Besides this, the MSK-impact dataset was filtered to include only breast cancers. Mutation and copy-number data for the BASIS dataset were obtained from Supplementary Tables 4, 14, and 20 accompanying reference[50] and filtered using similar criteria as the other datasets. The resulting datasets were merged and, where possible, annotated with the ER, PR, and HER2 status of the corresponding samples. To select for samples with deleterious missense mutations in *BRCA1*, *BRCA1* missense mutations were annotated for their expected pathogenicity using the Breast Cancer Information Core (BIC) database [http://research.nhgri.nih.gov/bic] and Align-GVGD [http://agvgd.iarc.fr]. We only selected samples with *BRCA1* missense mutations that were considered to be pathogenic (annotated as clinically important by BIC or Align-GVGD assigned class C15, C25, and C65). The final dataset was visualized using a custom script, focusing on cancer-associated genes, as defined by cBioPortal [http://cbioportal.org].

**RNA sequencing**. llumina TruSeq mRNA libraries were generated and sequenced with 50–65 base single reads on a HiSeq 2500 using V4 chemistry (Illumina Inc., San Diego)[50]. The resulting reads were trimmed using Cutadapt (version 1.15) to remove any remaining adapter sequences and to filter reads shorter than 20 bp after trimming to ensure good mappability. The trimmed reads were aligned to the GRCm38 reference genome using STAR (version 2.5.3a). QC statistics from Fastqc (version 0.11.5) and the above-mentioned tools were collected and summarized using Multiqc (version 1.1). Gene expression counts were generated by feature-Counts (version 1.5.2) using gene definitions from Ensembl GRCm38 version 89.

Normalized expression values were obtained by correcting for differences in sequencing depth between samples using DESeqs median-of-ratios approach and then log-transforming the normalized counts.

**Generation of CNA profiles and data analysis.** llumina HiSeq 2500 was performed using V4 chemistry (Illumina Inc., San Diego)[51]. The resulting reads were trimmed using Cutadapt (version 1.15) to remove any remaining adapter sequences and trim reads longer to a length of 50 bp for QDNAseq. Additionally, reads shorter than 20 bp after trimming were removed to ensure good mappability. The trimmed reads were aligned to the the GRCm38 reference genome using BWA aln (version 0.7.15). QC statistics from Fastqc (version 0.11.5) and the above-mentioned tools were collected and summarized using Multiqc (version 1.1). The resulting alignments were analyzed using QDNAseq (version 1.14.0) using the mm10 reference genome (with a 50 K bin size, 50 bp read lengths and default settings for other parameters) to generate copy-number logratios, segmented profiles and calls. The segmented profiles were analyzed using RUBIC (version 1.0.3; 6) to identify recurrent CNAs regions (focal threshold = 1e+08, min probes = 4 and FDR = 0.25). Genes with copy-number values were identified using a custom script, in which missing values were imputed from surrounding bins (with window size = 11, requiring at least 5 non-missing values). Copy-number instability was scored by calculating the fraction of bins with logratio values above/below a threshold of +/−0.5 in the segmented copy-number data.

**CNA analysis of BLBCs from TGCA.** Segmented copy-number data for the TCGA breast cancer samples were downloaded from firebrowse (version 2016_01_28). These data were matched to subtype annotations from TCGA and filtered for BLBC samples. This BLBC dataset was analyzed using RUBIC (version 1.0.3) to identify recurrent CNAs (with focal threshold = 1e+07, min probes = 260,000 and FDR = 0.25).

**Comparative oncogenomics analysis.** Candidate genes were initially selected by identifying mouse genes that were recurrently amplified/deleted in the RUBIC analysis of the combined germline/somatic Myc-driven mouse tumors. These candidates were subsequently annotated for their orthologous human genes using Ensembl Biomart (GRCH37) and filtered for candidates whose orthologues were also recurrently aberrated in the RUBIC analysis of the human BLBCs. To filter for correlation with expression, we calculated the Spearman correlation between copy-number calls and gene expression values of the remaining candidate genes and selected genes with an absolute correlation >0.2, resulting in a list of cross-species candidates.

**Mouse studies.** Myc murine cDNA was obtained from a cDNA clone (Clone 8861953, Source BioScience), sequence-verified and inserted as FseI-PmeI fragment into the Frt-invCag-IRES-Luc shuttle vector[21], resulting in Frt-invCag-Myc-IRES-Luc. Frt-invCag-Met-IRES-Luc and Frt-invCag-Cas9-IRES-Luc were described in[51] and[52]. Flp-mediated knockin of the shuttle vectors in the WapCre;Brca1F/F;Trp53F/F;Col1a1-frt GEMM-ESC was performed as described[21]. Chimeric animals were crossed with WB1P or B1P mice to generate the experimental cohorts. WapCre, Brca1F/F, Trp53F/F and knockin alleles were detected using PCR as described[20,21,53]. In vivo bioluminescence imaging was performed as described[51] by using a cooled CCD system (Xenogen Corp., CA, USA) coupled to Living Image acquisition and analysis software (Xenogen). Intraductal injections were performed as described[22,52]. Lentiviral titers ranging from 2–20 × 10^8 TU ml^−1 were used.

Orthotopic transplantation of WB1P and WB1P-Myc mammary tumors or organoids was performed by implanting small tumor fragments or cells into the fourth right mammary fat pad of nude mice as described previously[30]. Treatment was initiated when tumors reached a size of ~100 mm^3 (formula for tumor volume: 0.5 × length × width^2). Cisplatin (6 mg kg^−1 i.v.) was administered at day 0 and 14. Olaparib (100 mg kg^−1 i.p.) and AZD2461 (100 mg kg^−1 per os) were administered daily for 28 consecutive days. S63845 (25 mg kg^−1 i.v.) was administered once-weekly for 5 weeks[32]. For experiments with PDX-110[32], thawed single cell suspensions of the tumor were transplanted orthotopically into the fourth right mammary fat pad of NOD-SCID-IL2Rγc^−/− mice. Treatment was initiated when tumors reached a size of ~100 mm^3. Olaparib (50 mg kg^−1 i.p.) was administered 5 days out of 7 for 4 weeks. S63845 (25 mg kg^−1 i.v.) was administered once-weekly for 4 weeks. Vehicle was DMSO/10% (2-hydroxypropyl-b-cyclodextrin) for olaparib and 20% (2-hydroxypropyl-b-cyclodextrin)/HCl for S63845.

Animal experiments were approved by the Animal Ethics Committees of the Netherlands Cancer Institute and the Walter and Eliza Hall Institute of Medical Research. Mice were bred and maintained in accordance with institutional, national and European guidelines for Animal Care and Use.

**Histology and immunohistochemistry.** Tissues were formalin-fixed overnight and paraffin-embedded by routine procedures. Haematoxylin and eosin staining was performed as described[54]. Immunohistochemical stainings were processed as described[50,54]. For MYC and MCL1 immunohistochemistry, primary rabbit antibody anti-Myc (1:1000, Abcam ab32072) or anti-Mcl1 (1:1000, Cell Signaling 94296S) were used. All slides were digitally processed using the Aperio ScanScope (Aperio, Vista, CA, USA) and captured using ImageScope software version 12.0.0 (Aperio).

**Code availability.** The entire analysis of RNA-Seq and CNV-Seq data was implemented using Snakemake (version 4.3.1) and is freely available on GitHub [https://github.com/jrderuiter/snakemake-rnaseq and https://github.com/jrderuiter/snakemake-cnvseq]. Any additional scripts are available from the corresponding author upon reasonable request.

**Reporting summary.** Further information on experimental design is available in the Nature Research Reporting Summary linked to this article.

## Data availability
All sequence data that support the findings of this study are available in the European Nucleotide Archive under accession number PRJEB30443. Any additional data are available from the corresponding author upon reasonable request.

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

## Acknowledgements

We thank Sjors Kas, Eva Schut, Bastiaan Evers, Ben Morris, Renske de Korte-Grimmerink, Natalie Proost and Rebecca Theeuwsen for providing reagents, technical suggestions, and/or help with the experiments. We are grateful for excellent support from the NKI animal facility, RHPC computing facility, flow cytometry facility, animal pathology facility, transgenic facility, preclinical intervention unit, core facility molecular pathology and biobanking (CFMPB), and genomics core facility. PDX-110 was derived from a primary breast tumor provided by the Victorian Cancer Biobank (supported by the Victorian Government, Australia). This work was carried out on the Dutch national e-infrastructure with the support of SURF Cooperative (e-infra160136). Financial support was provided by the Oncode Institute, the Netherlands Organization for Scientific Research (NWO: Cancer Genomics Netherlands (CGCNL), Cancer Systems Biology Center (CSBC), Netherlands Genomics Initiative (NGI) Zenith 93512009 (J.J.), VICI 91814643 (J.J.)), the European Research Council (ERC Synergy project CombatCancer), a National Roadmap grant for Large-Scale Research Facilities from NWO (J.J.) and National Health and Medical Research Council (Australia) grants 1113133 (J.E.V. and G.J.L.), 1078730 (G.J.L.) and 1102742 (J.E.V.).

## Author contributions

Conceived the study: S.A., J.R.d.R., L.H., C.S.B., L.F.A.W., J.J. Designed and supervised the experiments: S.A., J.R.d.R., L.H., C.S.B., F.V., M.H.v.M., I.J.H., M.v.d.V., J.E.V., G.J.L., L.F.A.W., J.J. Performed the experiments: S.A., J.R.d.R., L.H., C.S.B., C.L., F.V., F.F., A.D., E.v.d.B., B.S., B.v.G., R.d.B., M.H.v.M. Analyzed and interpreted data: S.A., J.R.d.R., L.H., C.S.B., F.V., J.E.V., G.J.L., L.F.A.W., J.J. Wrote the paper: S.A., J.R.d.R., J.J.

## Additional information

**Competing interests:** The authors declare no competing interests.

